# Microecological Shifts in the Rhizosphere of Perennial Large Trees and Seedlings in Continuous Cropping of Poplar

**DOI:** 10.3390/microorganisms12010058

**Published:** 2023-12-28

**Authors:** Junkang Sui, Chenyu Li, Yinping Wang, Xiangyu Li, Rui Liu, Xuewen Hua, Xunli Liu, Hui Qi

**Affiliations:** 1College of Agronomy and Agricultural Engineering, Liaocheng University, Liaocheng 252000, China; lichenyu200209@163.com (C.L.); wyp352352@163.com (Y.W.); lxy1970668316@163.com (X.L.); lr08240717@163.com (R.L.); huaxuewen@lcu.edu.cn (X.H.); 2College of Forestry, Shandong Agricultural University, Tai’an 271000, China; xunliliu@163.com

**Keywords:** poplar, rhizosphere, continuous cropping obstacle, microbial community

## Abstract

The cultivation of poplar trees is hindered by persistent cropping challenges, resulting in reduced wood productivity and increased susceptibility to soil-borne diseases. These issues primarily arise from alterations in microbial structure and the infiltration of pathogenic fungi. To investigate the impact on soil fertility, we conducted an analysis using soil samples from both perennial poplar trees and three successive generations of continuously cropped poplar trees. The quantity and community composition of bacteria and fungi in the rhizosphere were assessed using the Illumina MiSeq platform. The objective of this study is to elucidate the impact of continuous cropping challenges on soil fertility and rhizosphere microorganisms in poplar trees, thereby establishing a theoretical foundation for investigating the mechanisms underlying these challenges. The study found that the total bacteria in the BT group is 0.42 times higher than the CK group, and the total fungi is 0.33 times lower than the CK group. The BT and CK groups presented relatively similar bacterial richness and diversity, while the indices showed a significant (*p* < 0.05) higher fungal richness and diversity in the CK group. The fractions of *Bacillus* were 2.22% and 2.41% in the BT and CK groups, respectively. There was a 35.29% fraction of *Inocybe* in the BT group, whereas this was barely observed in the CK group. The fractions of *Geopora* were 26.25% and 5.99%, respectively in the BT and CK groups. Modifying the microbial community structure in soil subjected to continuous cropping is deemed as the most effective approach to mitigate the challenges associated with this agricultural practice.

## 1. Introduction

The escalating issues of resource deficiencies and environmental degradation have prompted a heightened emphasis on forests globally. Populus plants, characterized as adaptable and semi-evergreen forest trees, exhibit a broad distribution in northern China [1]. Due to its rapid growth rate, timber yield, coppicing capability, and adaptability to diverse environments, poplar has emerged as a favored tree species for bioenergy plantations in temperate climates [2,3]. The poplar tree, belonging to the Populus spp., holds significant importance as a timber forest species globally. However, the practice of short-term rotating harvests and whole-tree harvesting of poplar has resulted in detrimental consequences such as severe site degradation and declining productivity [4]. At the same time, the prevalence of single cultivation and continuous cropping has become more widespread, significantly constraining wood production and ecological advantages [5]. The long-term practice of continuous cropping has a negative impact on the sustainability of agriculture due to the deterioration of soil texture, increased prevalence of crop diseases, and reduced crop yields [6]. This degradation of the soil environment and its detrimental effect on plant health and productivity present significant challenges to agricultural production [7].

Multiple factors contribute to the aforementioned obstacles, encompassing soil erosion, soil-borne diseases, nutrient imbalances, alterations in soil microbial communities, and the presence of autotoxic substances [8]. The principal factors that contribute to the enduring obstacles can be concisely summarized as follows: a decline in the physicochemical properties of the soil, the accumulation of crop allelopathic substances (specifically phenolic acids), and modifications in the microbial community structure of the soil, all of which signify an imbalance in the microecological environment of the soil [9]. Microorganisms play a pivotal role in soil ecosystems through their facilitation of nutrient transformation, enhancement of soil structure, and degradation of harmful substances [10,11]. Additionally, they significantly contribute to the mitigation of soil-borne diseases [12]. With the development of sequencing technology, it has greatly facilitated the rapid, accurate, scientific, and comprehensive investigation of microbial community structure by employing high-throughput sequencing techniques [13].

The rhizosphere, enhancing the most types and quantities of microorganisms and the most active metabolic activities, encompassing reciprocal interactions among plants, soil, and the microbiome, engages in frequent exchanges of materials and signals, thereby establishing rhizosphere immunity as a defense mechanism against pathogenic assaults [14,15]. Multiple studies have demonstrated that continuous cropping has a significant impact on the composition of the microbial community in the rhizosphere, subsequently leading to the degradation of soil health in areas where continuous cropping is practiced [16,17]. The role of the soil’s bacterial population is vital in promoting the decomposition of organic matter, nutrient cycling, suppression of soil-borne diseases, and improvement in plant growth [18]. Studies have demonstrated notable variations in soil bacterial communities across different durations of continuous cropping [19]. In light of advancements in high-throughput sequencing technologies, researchers are currently directing their attention towards investigating the intricate relationship between soil microbes and continuous cropping practices [20,21]. It is believed that enhancing the structure and functionality of rhizosphere microorganisms may offer potential solutions to mitigate the challenges associated with continuous cropping [22].

In addition to forced fallow, implementing a well-planned crop rotation system proves to be an effective strategy for managing soil diseases. However, adopting crop rotation in facility agriculture leads to a reduction in both profit margins and investment risks for farmers [23]. In an effort to mitigate degradation of soil fertility and growth of soil-borne diseases, farmers resort to the increased utilization of chemical fertilizers and pesticides, thereby incurring elevated expenses and generating excessive pesticide residues in crops [24]. Consequently, the excessive application of chemical fertilizers has given rise to significant environmental concerns, including soil compaction and nutrient imbalances [25]. This study aimed to investigate the microbial community structure and soil physicochemical properties of the rhizosphere soil in continuous cropping soil, focusing on perennial trees and newly planted poplar seedlings. The objective was to assess the influence of changes in rhizosphere microecology on continuous cropping obstacles.

## 2. Materials and Methods

### 2.1. Study Site and Experimental Design

The study site (31°56′ N, 117°08′ E) is located in Tai’an City, Shandong Province, in eastern China. Despite being close to Dawen River, the study site has not encountered any instances of flooding in recent years. An average growing season of 195 frost-free days and a temperate monsoon climate characterize this region. Temperatures range from −20.7% to 38.1%, with the average being 12.9 °C. The annual precipitation, occurring between July and September, amounts to approximately 697 mm. Climate information is sourced from the Statistics Bureau of Tai’an City.

The experiment was conducted within an artificial forest spanning approximately 3.5 km^2^. An experiment area measuring 12 m by 80 m was oriented north to south and rectangular in shape. The study employed a block design within the artificial forest. Each group area comprised three parallel rows, with seedlings planted at two-meter intervals within the rows and rows spaced 4 m apart. The perennial big trees (BTs) had been growing for over a decade without consecutive cropping, while the new seedling poplar trees (CKs) were spaced 20 m apart. The new seedling poplar trees were fertilized with 5–8 kg of compound fertilizer around their roots, twice a year, in the spring and autumn seasons.

The planting of poplar seedlings at the designated location spanned three successive generations. On 30 March 2018, seedlings of comparable height were replanted after three consecutive cycles of cropping without rotation.

### 2.2. Sampling and Counting Culturable Microorganisms

On 30 October 2020, the sampling procedure was carried out, wherein a random selection of five trees was made from the north, south, and central locations. The roots of these trees were subjected to sampling, whereby any surplus bulk soil was eliminated, and the soil adhering to the roots was identified as the rhizosphere soil. Subsequently, soil samples of the rhizosphere soil were collected at a depth ranging from 15 to 20 cm from the south, central, and north locations. These samples were then combined to form three triplicate soil samples for each location and treatment. The soil samples were individually collected and stored in sterile plastic bags, followed by transportation to our laboratory on ice. Subsequently, they were maintained at a temperature of −80 °C until DNA isolation. To determine the final samples for analysis, replicate samples were pooled and a quartile approach was employed.

The soil samples underwent serial dilution, reaching a dilution factor of 10-6, and the resulting dilutions were then inoculated onto nutrient agar medium for bacterial counting or onto Martin agar medium supplemented with 30 g/mL streptomycin for fungal counting. Subsequently, the agar plates were incubated at a temperature of 28 ± 2 °C for a duration of 2–3 days [26].

### 2.3. Edaphic Properties Determination in Rhizosphere Soil

Soil organic carbon (SOC) was measured utilizing the Walkley–Black method [27]. A solution comprising 10 mL 1 N potassium dichromate and 20 mL concentrated H2SO4 was amalgamated with 0.1 g of sieved and desiccated soil, followed by a gentle rotation for a duration of one minute. The resultant mixture was subsequently incubated at a temperature of 150 °C for a duration of ten minutes, after which it was allowed to cool down to room temperature. Subsequently, the samples were diluted to a final volume of 200 mL using deionized H2O, and 10 mL of H₃PO₄, 0.2 g NH4F, and 10 drops of (C₆H₅)₂NH indicator were added. Subsequently, the excess dichromate was titrated using Morh salt solution (0.5 N FeNH_4_SO_4_ and 0.1 NH_2_SO_4_) to determine the available P (AP) through the molybdenum blue method [28]. The spectrophotometer (UV2550, Shimadzu, Kyoto, Japan) was utilized to conduct the analysis. The available K (AK) was determined through soil extraction with C₂H₇NO₂. The quantification of total N (TN) was carried out using an automatic Kjeldahl distillation–titration unit (Foss, Hillerød, Sweden). The pH of the soil samples was measured using a pH meter (Mettler Toledo, Zürich, Switzerland).

### 2.4. Isolation and PCR Amplification of DNA

The microbial DNA from both groups was extracted utilizing the E.Z.N.A. Soil DNA Kit (Cat No, Omega Bio-tek, Norcross, GA, USA). Subsequently, the PCR assay was conducted targeting the V3–V4 region of the bacterial 16S ribosomal RNA, with the following cycling conditions: an initial denaturation at 95 °C for three minutes, followed by 27 cycles of denaturation at 95 °C for 30 s, annealing at 55 °C for 30 s, and extension at 72 °C for 45 s. A final extension step was carried out at 72 °C for ten minutes. The primers used were 338F (5′-barcode-ACTCCTACGGGAGGCAGCA-3′) and 806R (5′-GGACTACHVGGGTWTCTAAT-3′) [29], where the barcode represents a unique eight-nucleotide sequence for each sample. The polymerase chain reaction (PCR) was conducted in triplicate, utilizing 20-μL reaction mixtures comprising 5× FastPfu Buffer (4 μL), 2.5 mM dNTPs (2 μL), each primer (5 μM; 0.8 μL), FastPfu Polymerase (0.4 μL), 10 ng template DNA, and ddH2O to attain a final volume of 20 µL.

The PCR conditions for amplifying the fungal ITS1 region were as follows: an initial denaturation step at 95 °C for 3 min, followed by 35 cycles of denaturation at 95 °C for 30 s, annealing at 55 °C for 30 s, and extension at 72 °C for 45 s, with a final extension step at 72 °C for 10 min. The primers used for amplification were ITS1F (5′-barcode-CTTGGTCATTTAGAGGAAGTAA-3′) and 2043R (5′-GCTGCGTTCTTCATCGATGC-3′) [30], where the barcode comprising an eight-nucleotide sequence that is distinct for each sample, was utilized. The PCR procedure was conducted in triplicate, employing 20-μL reaction mixtures as previously described.

### 2.5. Real-Time (q)PCR

The oligonucleotide primers employed for the amplification of the V3–V4 region of bacterial 16S ribosomal RNA and the ITS1 region of fungi are delineated in Section 2.3. The reaction mixture for quantitative polymerase chain reaction (qPCR) comprised each primer at a concentration of 10 μM (0.5 μL), 25 μL of 2× SYBR Green qPCR Master Mix (12.5 μL), template DNA (2 μL), and ddH_2_O (9.5 μL). The amplification conditions involved a melting-curve analysis and a temperature profile of 95 °C for 10 min, followed by 40 cycles of 95 °C for 15 s and 60 °C for 1 min. An ABI 7500 fluorescence quantitative analyzer was used for data analysis, with a baseline starting at 3 and ending between 11 and 12.

### 2.6. Illumina MiSeq Sequencing

The purification of amplicons from 2% agarose gels was conducted using the AxyPrep DNA Gel Extraction Kit (Cat No, Axygen Biosciences, Union City, CA, USA), following the manufacturer’s instructions. Subsequently, the resulting products were quantified using the QuantiFluor-ST Kit (Cat No, Promega, Madison, WI, USA). The amplicons were then pooled in equimolar amounts for paired-end sequencing (2 × 250) on an Illumina MiSeq platform, following the manufacturer’s protocol.

### 2.7. Processing of Sequencing Data

To eliminate chimera sequences and acquire sequences of superior quality for subsequent data analysis, the UCHIME function within the Mothur (version 1.31.2) software was employed. The initial step involved the demultiplexing of raw fastq files, followed by quality filtering using QIIME (version 1.9.1) in subsequent stages. Firstly, a sliding window of 50 bp was established, and if the average mass value within the window was determined to be below 20, all sequences from the posterior end of the base to the anterior end of the window were excised. Additionally, sequences with a length of less than 50 bp after quality control were eliminated. Secondly, the overlap sequence was spliced based on the specified overlap base overlap criteria, with a maximum mismatch rate of 0.2 and an overlap length exceeding 10 bp. Any sequences that could not be successfully stitched together were subsequently removed. The sequences were segregated based on their respective barcodes and primers located at both ends of the sequence. Sequences containing ambiguous bases were eliminated, with the exception of two base mismatches allowed for primers [31].

The clustering of operational taxonomic units (OTUs) was performed using UPARSE (version 7.1) http://drive5.com/uparse/ (accessed on 21 January 2021) [32,33] with a 97% similarity cutoff. Chimeric sequences were identified and removed using UCHIME [34]. The taxonomy of each 16S rRNA gene was analyzed against the Silva (SSU128) 16S rRNA data resource using the RDP Classifier http://rdp.cme.msu.edu/ (accessed on 30 January 2021) with a confidence threshold of 70% [35]. Similarly, each ITS sequence was taxonomically assessed through the UNITE/ITS data resource using the RDP Classifier with a confidence threshold of 70% [36].

### 2.8. Statistical Analyses

The mean ± standard deviation (SD) was used to present all results. Significant differences in the quantities of culturable and total microorganisms, as well as diversity and richness indices, were observed between the BT and CK groups. With a significance level of *p* < 0.05, we used ANOVA and Duncan’s multiple range tests to investigate differences between groups. All statistical analyses were performed using SAS 9 (SAS Institute Inc., Cary, NC, USA).

## 3. Results

### 3.1. Changes in Edaphic Properties from March 2018 to October 2020

In March 2018, poplar seedlings were planted, and the soil’s organic carbon (OC), available phosphorus (AP), available potassium (AK), and total nitrogen (TN) exhibited significant differences between the BT group and CK group (Table 1). Specifically, the OC content in the BT group was significantly higher than that in the CK group, while the concentrations of AP, AK, and TN in the soil were significantly lower in the BT group compared to the CK group. Notably, there were no significant changes in soil properties within the BT group from March 2018 to October 2020. Furthermore, the differences between the BT and CK groups observed in October 2020 remained consistent with those observed in 2018. Interestingly, the OC content in the CK group significantly increased, while the pH value decreased when comparing from October 2020 to 2018.

### 3.2. Differences in the Quantities of Microbes in Rhizosphere Soils of Poplars

Perennial poplar trees exhibit a significantly higher abundance of culturable and total bacteria in rhizosphere in comparison to continuous cropping poplar seedlings. In contrast, the CK group demonstrates a notable presence of culturable and total fungi compared to the BT group (Table 2). However, it is important to note that the BT rhizosphere harbors 51.50% of the culturable fungi and 3.39 times the amount of culturable bacteria when compared to the CK group. qPCR analysis further reveals that the BT rhizosphere contains 42.45% more quantities of bacteria than the CK rhizosphere, but 49.75% fewer quantities of fungi than the CK group, as shown in Table 3.

### 3.3. Sequencing Quality Evaluation

Based on the sequencing data, 49,207 and 50,480 bacterial 16S rDNA sequences were obtained for the BT and CK groups, respectively, as well as 63,079 and 68,753 fungal ITS sequences for the BT and CK groups. Utilizing a clustering dissimilarity cutoff of 3%, the reads were categorized into various operational taxonomic units (OTUs). The Sobs diversity rarefaction curves for both bacterial and fungal communities did not reach an asymptote at a distance level of 0.03 (Figure 1), indicating that not all communities present in the samples were adequately represented by the sequencing data. Nevertheless, the Shannon diversity index and rarefaction curves were integrated to obtain a more comprehensive picture of diversity within communities (Figure 2). According to the Shannon diversity curves, as the number of reads increased, the curves plateaued, indicating that sufficient data were collected to examine communities.

The analysis of diversity and richness indices in the soil samples (Table 3) revealed that the BT and CK groups exhibited similar levels of bacterial richness and diversity. However, the ACE and Chao values, which serve as indicators of species richness, were relatively higher for the rhizosphere bacterial community in the CK group. The Shannon and Simpson diversity indices displayed a consistent pattern, whereby greater Simpson values were associated with lower diversity index values.

In terms of fungal communities, the ACE and Chao values in the BT group exhibited a substantial decrease compared to those in the CK group. Furthermore, the Shannon diversity index in the BT group showed a significant reduction in comparison to the CK group. The higher Simpson index observed in the BT group indicated a lower level of diversity. The results of this study offer empirical support for the notion that the CK group exhibited a greater abundance and diversity of fungal communities when compared to the BT group. Conversely, there were only negligible disparities observed in the diversity and species richness of bacterial communities between the BT and CK groups.

### 3.4. Differences in Community Composition and Structure

The Mothur program was employed for sequence classification. In the BT and CK groups, the Actinobacteria phylum constituted 26.67% and 30.21%, respectively, followed by Proteobacteria with 18.60% and 18.29% in the BT and CK groups, respectively, and Acidobacteria with 17.99% and 16.70% in the BT and CK groups, respectively. Regarding the dominant fungal phyla, Ascomycota accounted for 51.52% and 63.76% in the BT and CK groups, respectively, while Basidiomycota represented 39.84% and 21.78% in the BT and CK groups, respectively (Figure 3a,b). The results of the heatmap analysis provided confirmation that Actinobacteria, Proteobacteria, and Acidobacteria were the prevailing bacterial phyla, whereas Ascomycota and Basidiomycota emerged as the dominant fungal phyla (Figure 4a,c).

The bacterial composition exhibited overall similarity between both groups, although there were notable discrepancies in the distribution of individual genera (Figure 3). Specifically, Enterococcus comprised 8.34% of the total classifiable bacterial sequences in the BT group, while accounting for 9.21% in the CK group (Figure 3b). The fractions of Bacillus were 2.22% and 2.41% in the BT and CK groups, respectively. Notably, the proportion of Streptomyces in the BT group was significantly lower at 0.73% compared to the CK group at 1.37%. Similarly, the *Nocardioides* proportion in the BT group was lower at 0.67% compared to the CK group at 1.34%. Additionally, the fraction of *Roseiflexus* in the BT group was significantly lower at 0.61% compared to the CK group at 1.54%. In contrast, the fraction of *Nitrospira* in the BT group was significantly higher (2.26%) compared to the CK group (1.43%). Additionally, the fraction of *Gaiella* in the BT group was significantly higher (2.31%) relative to the CK group (1.95%). The heatmap analysis provided insights into the quantitative relationships between fungi and bacteria in the BT and CK groups. In the BT group, the fraction of Enterococcus was 8.34%, whereas in the CK group, it was 9.22%, indicating the largest disparity in bacterial composition between the two groups. Conversely, the proportion of *Blastococcus* was 0.44% and 1.31% in the BT and CK groups, respectively. Similarly, the percentages of *Nitrospira* were 2.26% and 1.43%, and those of *Anthrobacter* were 0.54% and 1.28% in the BT and CK groups, respectively (Figure 4b).

In the BT group, the fraction of *Inocybe* was 35.29%, whereas it was scarcely observed in the CK group. The fractions of *Geopora* were 26.25% and 5.99% in the BT and CK groups, respectively. Additionally, the fraction of *Fusarium* in the BT group was 6.94%, which exhibited a significant increase compared to the CK group’s fraction of 4.28%. *Ceratobasidium* accounted for 1.07% in the BT group, which was remarkably higher than the CK group’s fraction of 0.15%. Conversely, the fraction of *Mortierella* was 5.77% and 7.52% in the BT and CK groups, respectively. *Cryptococcus* constituted 0.87% of the BT group, whereas it exhibited a significantly higher prevalence of 5.80% in the CK group. *Guehomyces* accounted for 0.21% in the BT group, which was notably lower than its occurrence of 3.64% in the CK group. *Preussia* constituted 0.11% in the BT group, which was remarkably lower than its prevalence of 2.56% in the CK group. Similarly, *Gibberella* constituted 0.62% in the BT group, which was significantly lower than its occurrence of 1.93% in the CK group. In terms of fungal composition, the heatmap analysis revealed that the fractions of *Inocybe* in the BT and CK groups were 35.30% and 0.01%, respectively. Similarly, the fractions of *Geopora* in the BT and CK groups were 26.26% and 5.99%, respectively. The fractions of *Fusarium* species were 6.94% and 4.28% in the BT and CK groups, respectively, while the fractions of *Motierella* were 5.77% and 7.52% in the BT and CK groups, respectively (Figure 4d).

The Venn diagram in Figure 5 illustrates the presence of bacterial and fungal operational taxonomic units (OTUs) in both study groups. At a dissimilarity level of 3%, the number of bacterial OTUs observed in the BT treatment was 2310, while the CK treatment had 2505 OTUs. Among these, 2176 OTUs were shared between the BT and CK groups. Notably, three primary genera (*Enterococcus*, *Bacillus*, and *Acidothermus*) were found in both groups, while *Chryseobacterium* and *Nitrosomonas* were exclusively present in the BT group. Conversely, *Angustibacter*, *Lactobacillus*, and *Leptolyngbya* were only detected in the CK group.

A total of 1111 fungal operational taxonomic units (OTUs) were identified in the BT group, while the CK group exhibited 1343 OTUs, both at a dissimilarity level of 3%. Among these, 743 OTUs were shared between the two groups, with *Geopora*, *Mortierella*, *Fusarium*, and *Cryptococcus* being the common fungal taxa. Conversely, *Abortiporus*, *Clavulina*, *Endosporium*, and *Melanoleuca* were exclusively detected in the BT group, whereas *Asterotremella*, *Bulleromyces*, *Calcarisporiella*, and *Helicobasidium* were solely observed in the CK group.

The present study employed LEfSe analysis to identify microbial genera in the rhizosphere of continuous cropping poplar seedlings and perennial poplar. The results revealed that the BT group exhibited specific bacterial genera, namely, *Haliangium*, *Phytohabitans*, *Hamadaea*, *Flavobacterium*, and *Asteroleplasma*. Conversely, the CK group displayed specific bacterial genera, including *Roseiflexus*, *Arthrobacter*, *Streptomyces*, *Microvirga*, and *Lechevalieria*. Furthermore, the BT group exhibited specific fungal genera, such as *Inocybe*, *Geopora*, *Pluteus*, *Abortiporus*, and *Mycothermus*, while the CK group demonstrated *Cryptococcus*, *Guehomyces*, *Gibberella*, *Sporobolomyces*, and *Pyrenochaetopsis* (Figure 6).

## 4. Discussion

The increasing prevalence of intensive cultivation and monoculture cropping practices has contributed to a rise in the incidence of replant diseases. The continuous cultivation of crops in a single plot over an extended period of time can result in the deterioration of crop growth conditions, decreased yield and quality, and the exacerbation of diseases [37]. Several factors have been identified as significant contributors to the ongoing difficulties associated with continuous cropping, including soil-borne diseases, imbalances in nutrient levels, and changes in the physicochemical characteristics of the soil, such as shifts in bacterial and fungal communities [8]. The high mortality rate of plants is believed to be a direct result of the challenges posed by continuous cropping [38], primarily due to disruption of the rhizosphere microorganisms [39]. Soil microbial communities have been found to be significantly altered by continuous cropping over time, resulting in an increased proliferation of various taxa within this community [40].

When comparing the microbial quantity between the BT group and the CK group, it was observed that the number of culturable and total bacteria in the BT group was significantly higher than that in the CK group. Conversely, the number of culturable and total fungi in the BT group was significantly lower than that in the CK group. Additionally, the BT group exhibited higher organic carbon (OC) levels, but lower levels of available phosphorus (AP), available potassium (AK), and total nitrogen (TN) compared to the CK group. This phenomenon can be attributed to the persistent application of chemical fertilizers in consecutive cropping systems, which has led to the prevalence of bacteria in the BT group with reduced fertility. These results are similar to previous studies where there were less bacterial populations in chemical fertilizers treatment [41]. A prior study hypothesized that the decrease in beneficial microbial populations and the rise of pathogenic microorganisms in the soil contribute to the displacement of rhizosphere soil microflora and bacteria by less fertile fungi [42]. The study found that the culturable bacterial contents in the BT group is 2.38 times higher than the CK group and the culturable fungal contents in the CK group is 0.94 times higher than the CK group. The total bacteria in the BT group is 0.42 times higher than the CK group, and the total fungi in the CK group is 0.50 times higher than the BT group. The abundance of fungi increased in the CK group than the BT group. There is a better bacteria/fungi ration in the BT group. This finding aligns with prior research indicating that a decrease in the density of beneficial microbial populations in the soil, coupled with an increase in pathogenic microorganisms, leads to a transition in the rhizosphere soil microflora from a high-fertility “bacterial type” to a low-fertility “fungal type” [43,44,45]. As a result, the long-term implementation of continuous cropping practices is likely to cause a decline in the diversity of the soil microbial community and an increase in harmful microbial populations [46]. Similar results were observed in the continuous cropping of cotton, which had an impact on the diversity and structure of bacterial communities. The decline of *Nocardia* and *Streptomyces* in Actinomycetes may contribute to the challenges associated with continuous cropping [6].

Furthermore, the comparison of bacterial diversity and richness indexes between the BT group and CK group revealed that the BT group exhibited similar bacterial community abundance, but significantly lower fungal community abundance compared to the CK group. The Actinobacteria, Proteobacteria, and Acidobacteria phyla have been identified as the dominant bacterial groups in soil ecosystems, and numerous studies have emphasized their importance as antagonistic microorganisms. Acidobacteria, in particular, are found extensively in soil environments [47]. Moreover, research has shown their participation in the degradation of polysaccharides from plants and microorganisms, as well as their association with soil nitrogen availability [48]. The sequencing analysis demonstrated that Enterococcus was the prevailing genus in both the BT and CK rhizospheres, aligning with its recognized abundance in these environments. Notably, the BT group exhibited substantially higher proportions of *Nitrospira* and *Gaiella* compared to the CK group. Conversely, the BT group displayed significantly lower proportions of Streptomyces, *Nocardioides*, and *Roseiflexus* in comparison to the CK group. These findings are consistent with prior investigations [49,50,51]. Members of *Streptomyces* have attracted attention due to them being an important source of natural commercial products such as biotics [52]. They have been found to be widely distributed in various habitats [53,54]. A variety of *Nocardioides* and *Mortierella* strains play an important role in bioremediation of contaminated regions by degrading organic pollutants [55,56]. Moreover, there is increasing evidence that these strains can oxidize inorganic compound and carbon monoxide [57]. Members of the genus of the order Mortierellales within the family Mortierellaceae are filamentous fungi commonly found in soil. The genus contains a large number (nearly 100) of validated species that can be found on almost any substrate and is often encountered in soils as saprophytes [58].

Furthermore, most of the fungi in both sample sets belonged to the Ascomycota phylum. Previous research has emphasized the importance of Ascomycota in soil ecosystems due to its ubiquitous distribution and prevalence in soils under continuous cropping [59,60]. Notably, the proportion of *Fusarium* in the BT group was significantly elevated (6.94%) compared to the CK group (4.28%). This situation might be caused by the presence of litter from perennial poplar forests soil [61]. The CK group displayed a higher level of diversity in the composition of the fungal community. Concurrently, the CK group exhibited a 5.82% occurrence of *Rhizoctonia*, indicating the emergence of distinct fungal pathogenic communities in soils subjected to continuous cropping, which aligns with prior studies [62,63].

By utilizing high-throughput sequencing techniques to assess alterations in microbial community structure, our analysis primarily focused on evaluating the abundance of 24 major bacterial genera and 20 major fungal genera within the soil of perennial poplar trees, as well as in soils subjected to continuous cropping for three generations of poplar trees. Our findings indicate that fungal diversity and richness were higher in continuous cropping soils compared to perennial tree soils. Specifically, we observed significant increases in the abundance of *Streptomyces*, *Nocardioides*, and *Roseiflexus* in continuous cropping soil. However, there is a lack of studies investigating these specific fungal species, and further research is needed to better understand their characteristics and roles in continuous cropping systems. Additionally, the presence of *Mortierella* and *Cryptococcus* was observed to increase the continuous cropping soil, but their functional mechanisms require further investigation.

## Figures and Tables

**Figure 1 microorganisms-12-00058-f001:**
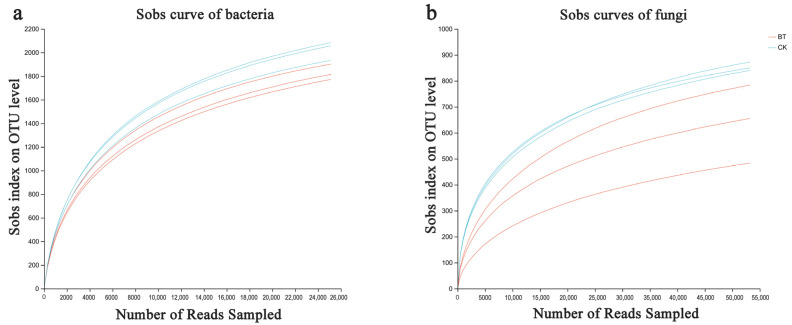
Bacterial (**a**) and fungal sobs curves (**b**) were examined to assess the impact of a 3% dissimilarity cutoff on the identification of uncovered operational taxonomic units (OTUs). The abbreviation “BT” refers to perennial poplar big trees, while “CK” denotes replanting poplar seedlings in continuous cropping soil. Each treatment was subjected to analysis in triplicate.

**Figure 2 microorganisms-12-00058-f002:**
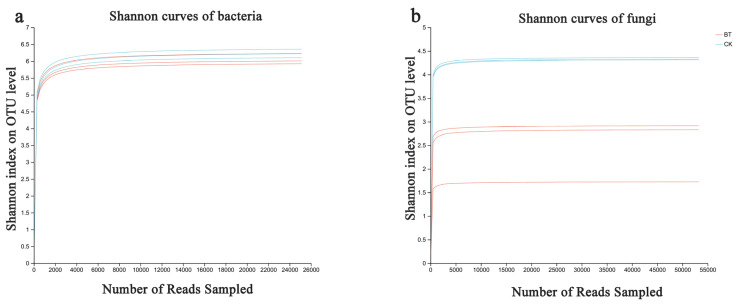
Bacterial and fungal Shannon curves (**a**,**b**) were examined to assess the impact of a 3% dissimilarity cutoff on the identification of uncovered operational taxonomic units (OTUs). The abbreviation “BT” refers to perennial poplar big trees, while “CK” denotes the replanting of poplar seedlings in soil with a history of continuous cropping. Each treatment was subjected to analysis in triplicate.

**Figure 3 microorganisms-12-00058-f003:**
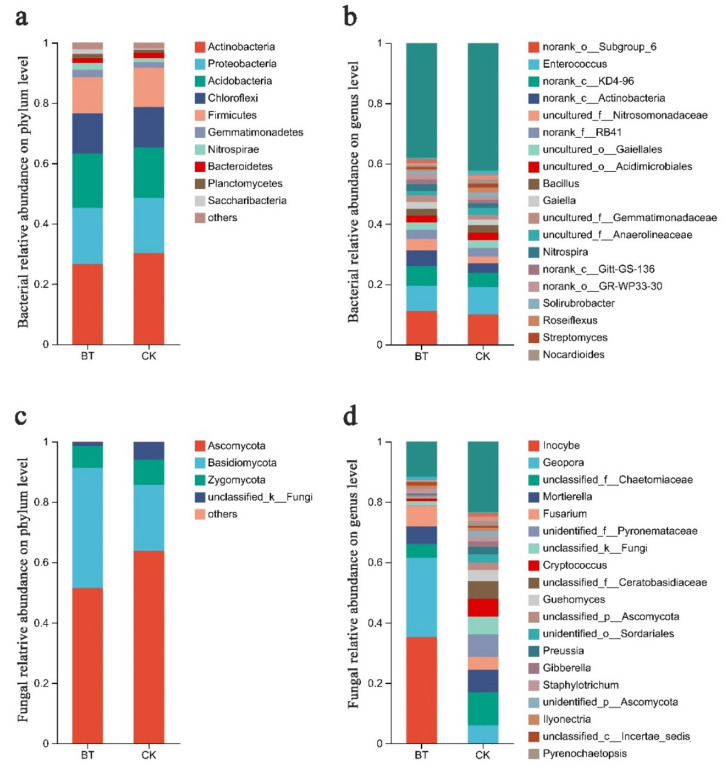
Communities of bacteria and fungi in the study group. (**a**) Relative abundances of bacteria at the phylum level; (**b**) relative abundances of bacteria at the genus level; (**c**) relative abundances of fungi at the phylum level; (**d**) relative abundances of fungi at the genus level. The relative abundances of major genera are illustrated in stacked bar graphs. The abbreviations “BT” and “CK” were used to denote perennial poplar trees and replanting poplar seedlings in continuous cropping soil, respectively. Each treatment was analyzed using three replicates.

**Figure 4 microorganisms-12-00058-f004:**
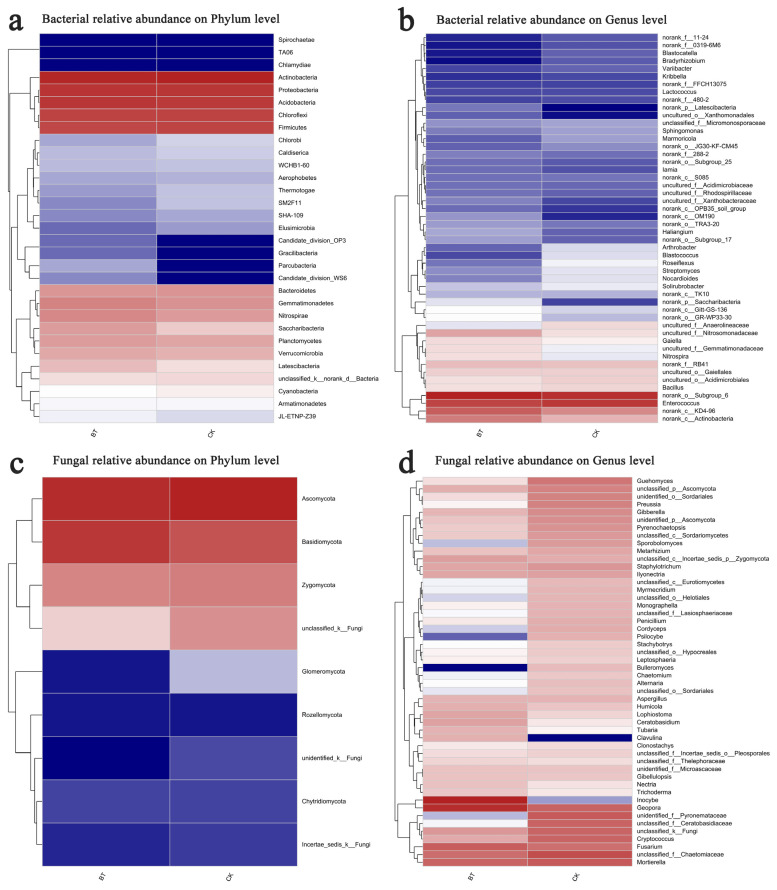
Hierarchical clustering of bacterial and fungal distributions. (**a**) Relative abundances of bacteria at the phylum level; (**b**) relative abundances of bacteria at the genus level; (**c**) relative abundances of fungi at the phylum level; (**d**) relative abundances of fungi at the genus level. Color gradient of color blocks to display the abundance changes of different species in the sample The abbreviations “BT” and “CK” were used to denote perennial poplar trees and replanting poplar seedlings in continuous cropping soil, respectively. Each treatment was analyzed using three replicates.

**Figure 5 microorganisms-12-00058-f005:**
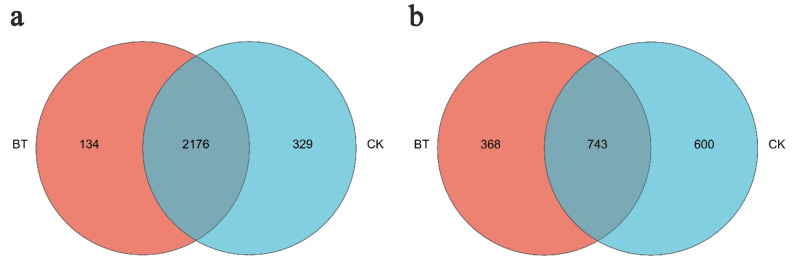
Unique and shared OTUs of (**a**) bacteria and (**b**) fungi for the two groups in Venn diagram. “BT” means perennial poplar trees, while “CK” means replanting poplar seedlings in continuous cropping soil. We analyzed three replicates for every treatment.

**Figure 6 microorganisms-12-00058-f006:**
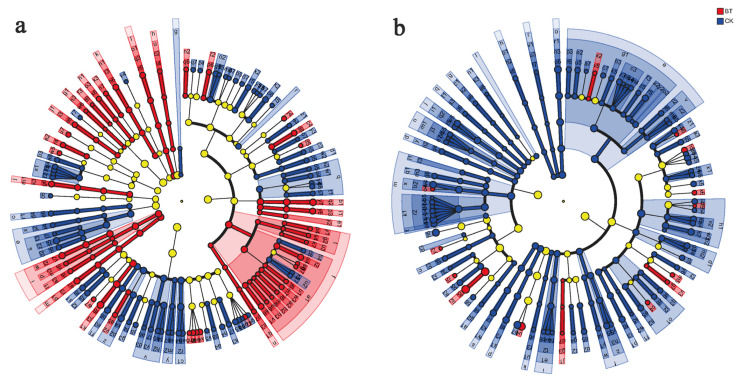
Discriminant analysis of muti-level species differences through LEfSe analysis. (**a**) Differences of bacterial multi-level species in the BT and CK groups. (**b**) Differences of fungal multi-level species in the BT and CK groups. Different colored nodes represent microbial communities that are significantly enriched in their corresponding groups and have a significant impact on inter group differences.

**Table 1 microorganisms-12-00058-t001:** Edaphic properties in perennial poplar big trees and poplar seedlings in soil with continuous cropping.

		Organic Carbon (g/kg)	Available P (mg/kg)	Available K (mg/kg)	Total N (mg/kg)	pH
March 2018	BT	7.41 ± 0.05 ^a^	23.31 ± 0.07 ^b^	121.24 ± 1.08 ^b^	695.31 ± 3.24 ^b^	7.21 ± 0.01 ^a^
	CK	6.05 ± 0.07 ^c^	35.63 ± 2.33 ^a^	132.28 ± 1.44 ^a^	765.82 ± 7.43 ^a^	7.23 ± 0.02 ^a^
October 2020	BT	7.35 ± 0.03 ^a^	24.07 ± 0.11 ^b^	125.31 ± 1.12 ^b^	701.23 ± 1.67 ^b^	7.19 ± 0.01 ^a^
	CK	6.27 ± 0.04 ^b^	35.38 ± 2.77 ^a^	139.20 ± 2.03 ^a^	791.58 ± 3.77 ^a^	7.07 ± 0.04 ^b^

Data are presented in the form of mean ± standard error (SE). A significant difference at *p* < 0.05 is indicated by lowercase superscript letters within the same column. The abbreviation “BT” refers to perennial poplar big trees, while “CK” represents the replanting of poplar seedlings in soil with continuous cropping.

**Table 2 microorganisms-12-00058-t002:** Amount of microorganisms in rhizosphere soils of the two groups.

Treatment	Quantity of Culturable Microbial		Quantity of Total Microbial	
	Bacterial × 10^7^ (cfu/g Soil)	Fungal × 10^6^ (cfu/g Soil)	Bacterial × 10^7^ (Copies/μL)	Fungal × 10^4^ (Copies/μL)
BT	3.25 ± 0.11 ^a^	4.76 ± 0.19 ^b^	1.01 ± 0.13 ^a^	1.97 ± 0.04 ^b^
CK	0.96 ± 0.17 ^b^	9.25 ± 0.49 ^a^	0.71 ± 0.02 ^b^	2.95 ± 0.10 ^a^

Data are presented in the form of mean ± standard error (SE). Lowercase superscript letters are used within the same column to indicate significant differences at *p* < 0.05. The abbreviation “BT” refers to perennial poplar big trees, while “CK” represents the replanting of poplar seedlings in soil with continuous cropping.

**Table 3 microorganisms-12-00058-t003:** The diversity and richness indices of bacterial and fungal communities in perennial poplar big trees and poplar seedlings in soil with continuous cropping.

	Sample	Cutoff	ACE	Chao	Shannon	Simpson	Coverage
Bacterial	BT	0.03	2186.33 ± 43.93 ^b^	2194.97 ± 42.72 ^a^	6.05 ± 0.16 ^a^	0.012 ± 0.027 ^a^	0.982177
	CK	0.03	2376.19 ± 100.38 ^a^	2378.34 ± 117.81 ^a^	6.22 ± 0.13 ^a^	0.012 ± 0.044 ^a^	0.981184
Fungal	BT	0.03	893.98 ± 65.89 ^a^	883.48 ± 46.29 ^b^	2.87 ± 0.52 ^b^	0.30 ± 0.147 ^a^	0.996437
	CK	0.03	1029.27 ±34.58 ^a^	1016.56 ± 27.96 ^a^	4.34 ± 0.02 ^a^	0.04 ± 0.005 ^b^	0.996280

Data are presented in the form of mean ± standard error (SE). A significant difference at *p* < 0.05 is indicated by lowercase superscript letters within the same column. The abbreviation “BT” refers to perennial poplar big trees, while “CK” represents the replanting of poplar seedlings in soil with continuous cropping.

## Data Availability

Most data are presented in the article and SRA database (Bioproject accession number: PRJNA913983).

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
