# Peer review of "Microecological Shifts in the Rhizosphere of Perennial Large Trees and Seedlings in Continuous Cropping of Poplar"

_microorganisms, 2023, doi:10.3390/microorganisms12010058_

Round 1

Reviewer 1 Report

Comments and Suggestions for Authors

The manuscript is focused on comparison of a microbial diversity in the rhizosphere of Populus spp., in dependence on the type of continuous cropping. The topic is important being of great economic value and representing significant data on the shift in the soil microbial community structure and potential functions under different agroforestry systems. The paper is well-written and interesting to read, it shows a broad range of up-to-date molecular methods and data analysis software.  Some remarks are listed below:

-        Authors formulated the aim, which is related to “establishing a theoretical foundation for investigating the mechanisms underlying the impact of continuous cropping and developing effective prevention and control strategies” (in Abstract), and “assessing the influence of rhizosphere microecology changes on continuous cropping obstacles and establish a theoretical foundation for overcoming these obstacles” (Introduction). However, the Discussion part does not focus on the developing effective prevention and control strategies in this context.

-        The differences in the abundance of bacterial genera in CK and BT (Page 9) were considered as significant, while it was just below one percent, e.g., “The proportion of Streptomyces in the BT group was significantly lower at 0.73% compared to the CK group at 1.37%”. The same is attributed to Nocardioides, Roseiflexus, Arthrobacter, Blastococcus, etc. and other description of results on fungal abundance in Figure 4d (page 10).

-        It would be important to compare the rhizosphere samples with unplanted soil.

-        It is advisable to give a feedback about the actual or theoretical spread of bacterial/fungal infections in poplar plantations.

-        The references to Table 1 and Table 2 in the text are not found.

Kind regards

Reviewer 2 Report

Comments and Suggestions for Authors

The manuscript addressed Rhizosphere Microecology in Continuous Cropping: Perennial Large Trees and Seedlings of Poplar. The following points need to be considered.

1- The title should be changed to (Microecological Shifts in the Rhizosphere of Perennial Large Trees and Seedlings in Continuous Cropping of Poplar). or (Microbes in the root zone: Comparing poplar trees and continuous crops).

2- The abstract lacks specific quantitative data and recommendations on practical strategies for addressing challenges in continuous cropping.

3- The introduction it lacks a coherent structure and focused organization. The comprehensive information provided feels disjointed and could be better organized for a smoother flow. Moreover, rationale and novelty should receive more stress.

4- The study's results may not be generalizable due to a limited sample size of three replicates per treatment, potentially underestimating the full diversity and variability of microbial communities. Additionally, the study focuses solely on dominant phyla and genera, neglecting the functional potential of the entire microbial community, hindering our understanding of each ecosystem's specific microbial roles.

5- The discussion lacks a deeper exploration of causative relationships. For instance, it mentions differences in soil properties between groups but doesn't sufficiently explain how these properties directly affect microbial populations.

6- A detailed comparison between microbial groups in the BT and CK groups was given but doesn't delve into the potential ecological implications of these variations. Further interpretations of how these differences might impact plant health, soil fertility, or disease dynamics could strengthen the discussion.

7- There's a lack of discussion on the functional roles of newly observed species like Streptomyces, Nocardioides, Roseiflexus, Mortierella, and Cryptococcus.

8- References should be revised ensuring scientific names are italic (for example reference 13 and 31)

Comments on the Quality of English Language

Minor editing of English language required

Reviewer 3 Report

Comments and Suggestions for Authors

The paper entitled “Microecological Changes in the Rhizosphere of Continuous Cropping Poplar - Perennial Large Trees and Continuous Cropping Seedlingsdescribes research directed to identify differences in the microbial community of perennial large trees and new seedling poplar trees.  Although interesting and valuable results are obtained, my main concern is the correlation between the set goal and the experimental design. Here I see two possibilities: First, the experimental design and treatments are not well described, and second, the objective does not correspond with the proposed trial. Comparing the microbial community between plans of different ages, in the absence of a detailed explanation, seems to be different from comparing the microbial community under different crop regimes.

For this reason, I recommend a major revision of the manuscript, and all other comments are included in the supplementary document.

Author Response

Response to Reviewer 3 Comments

  1. Summary

  1. Reviewer’s Evaluation

The paper entitled “Microecological Changes in the Rhizosphere of Continuous Cropping Poplar - Perennial Large Trees and Continuous Cropping Seedlings” describes research directed to identify differences in the microbial community of perennial large trees and new seedling poplar trees.  Although interesting and valuable results are obtained, my main concern is the correlation between the set goal and the experimental design. Here I see two possibilities: First, the experimental design and treatments are not well described, and second, the objective does not correspond with the proposed trial. Comparing the microbial community between plans of different ages, in the absence of a detailed explanation, seems to be different from comparing the microbial community under different crop regimes.

For this reason, I recommend a major revision of the manuscript, and all other comments are included in the supplementary document.

  1. Point-by-point response to Comments and Suggestions for Authors

Comments 1: Soil sickness should be explained or replaced with another term.

Response 1: Thank you very much for raising this question. We have realized that this statement is inappropriate and made changes of “degradation of soil fertility and growth of soil borne diseases”. At the same time, we also replied in the PDF version.

Comments 2: The aim of the research is not clear enough and do not corresponds with what was conducted. The paper describes differences between microbial community of big trees and new seedling, and not two different cropping systems. Changes in microbial community can be explained by different age of plants. With the aim defined like this, the experimental setup is not adequate.

Response 2: Thank you very much for raising this question. The growth period of poplar trees in the continuous cropping plot is about 5 years, and there is no growth period similar to that of large trees in the BT group. Due to the fact that poplar is a large woody plant, we chose to conduct experiments in the field. This experiment compared the differences in the rhizosphere microbial community structure between perennial trees and continuous cropping poplar seedlings in fields with similar distances. Indeed, as you mentioned, the changes in microbial community may be explained by the different ages of plants. However, it is very difficult to find continuous and non continuous field soils with the same soil fertility conditions, and the growth time of poplar trees is relatively long. Perhaps we should conduct pot experiments under laboratory conditions to verify the effect of planting systems on rhizosphere microorganisms. We have realized that this statement is inappropriate. In this way, We’ve made changes about the aim of the research, focused on the the influence of continuous cropping obstacles on rhizosphere microecology changes. We also replied in the PDF version.

Comments 3: Sources of climate information?

Response 3:  Thank you for pointing out the problem. Climate information is sourced from the Statistics Bureau of Tai'an City. We’ve added information in page 3, also replied in the PDF version.

Comments 4: As it was said in previous comments, the treatments are not clearly described, or they are not in consistent with the set aim of the study.

Response 4: Thank you again for pointing out the problem. We feel sorry for this. We’ve explained in Comments 2.

Comments 5: All the sampling refers to rhizosphere soil, or some bulk soil samples are taken? Not clear enough.

Response 5: Thank you very much for raising this question. We apologize for not providing accurate information. All the sampling refers to rhizosphere soil. We’ve added information in section 2.2, also replied in the PDF version.

Comments 6: Determination of Edaphic Properties in rhizosphere soil or in bulk soil?

Response 6: Thank you very much for raising this question. We apologize for not providing accurate information. It should be Determination of Edaphic Properties in rhizosphere soil. We’ve added information in section 2.3, also replied in the PDF version.

Comments 7: It is better not to use abbreviations in a table headings.

Response 7: Thank you again for pointing out the problem. We’ve made changes in Table 1, also replied in the PDF version.

Comments 8: Be more specific - "in rhizosphere" should be added in the first sentence

Response 8: Thank you again for pointing out the problem. We’ve made changes in Line 293, also replied in the PDF version.

Comments 9: The discussion of this part could be more developed, some explanations of the results given, as well as correlation with sequencing results.

Response 9: Thank you again for pointing out the problem. We’ve improved this in discussion part in page 13.

Comments 10: The headings should be more precize. Define "treatment".

Response 10: Thank you again for pointing out the problem. The “treatment” may not be appropriate, we’ve changed it into “group”.

Comments 11: There are only 3tables in the manuscript.

Response 11: Thank you for pointing out the problem. We feel sorry for this. It’s a miswriting, we’ve already changed it into Table 3.

Comments 12: The table title is missing.

Response 12: Thank you for pointing out the problem. We feel sorry for this. We’ve already added the table title.

Comments 13: The title should be changed or the text below should better correspond with the title. BT and CK were completely equally presented.

Response 13: Thank you for pointing out the problem. We’ve made changes of the title in page 9.

Comments 14: Again, it is not clear that here you compare two different practices. Additional explanation should be added about using differences in microbial communities of plants of different age to show the effects of cropping system.

Response 14: Thank you for pointing out the problem. According to the reviewer's comments, it is necessary to find two plots of land with similar soil fertility to compare the effects of different cultivation systems, including land after multiple generations of continuous planting of poplar trees and land without continuous planting of poplar trees, which is difficult to achieve in actual forest land. Another feasible method is to prepare continuous cropping multi generation soil under laboratory conditions for experimentation, but this method has not been implemented due to the larger individual size of poplar trees requiring more soil and their long growth years. Other explanations can be found in Comments 2.

Comments 15: This is not clear, because we do not have any data about fertilization in treatments. The experimental setup should be given in more details - otherwise it is not completely understandable.

Response 15: Thank you for pointing out the problem. We’ve added the fertilization treatments in page 3.

Comments 16: In that case, how do you explain better bacteria/fungi ration in older plants compared to CK?

Response 16: Thank you for pointing out the problem. In BT group, there is a better bacteria/fungi ration. This is consistent with previous report that suggested that when the density of beneficial microbial populations in the soil decreases and the number of pathogenic microorganisms increases, the rhizosphere soil microflora shifts from high-fertility “bacterial type” to low-fertility “fungal type”[1-3]. We’ve made explaination in page 13, and replied in the PDF version.

Comments 17: Previous research - which one?

Response 17: Thank you for raising this question. We feel sorry about it. It missed the references of 60 and 61, we’ve put them in the right place in page 13.

Comments 18: In the present study, or in the cited literature (49-50)?

Response 18: Thank you for pointing out the problem. The cited literature were in wrong position, we’ve put them in the right place. The data were in the present study.

  1. Wu, F.; Wang, X. Effect of Monocropping and Rotation on Soil Microbial Community Diversity and Cucumber Yield, Quality Under Protected Cultivation. Scientia Agricultura Sinica 2007, 555-561.
  2. Paudel, B.R.; Carpenter-Boggs, L.; Higgins, S. Influence of brassicaceous soil amendments on potentially beneficial and pathogenic soil microorganisms and seedling growth in Douglas-fir nurseries. Applied Soil Ecology 2016, 105, 91-100, doi:https://doi.org/10.1016/j.apsoil.2016.04.007.
  3. Patkowska, E. Biostimulants Managed Fungal Phytopathogens and Enhanced Activity of Beneficial Microorganisms in Rhizosphere of Scorzonera (Scorzonera hispanica L.). Agriculture 2021, 11, doi:10.3390/agriculture11040347.

Round 2

Reviewer 1 Report

Comments and Suggestions for Authors

The manuscript has been substantially improved. The pdf file with minor comments is attached. 

Kind regards

Author Response

Response to Reviewer 1 Comments

  1. Summary

  1. Reviewer’s Evaluation

The manuscript has been substantially improved.

  1. Point-by-point response to Comments and Suggestions for Authors

Comments 1: Hui Qi: affiliation is not indicated.

Response 1: Thank you very much for pointing out the problem. We’ve made it correct.

Comments 2: Higher or lower?

Response 2: Thank you very much for pointing out the problem. We apologize for not being able to express it clearly. We have made modifications in the corresponding positions and marked them in the PDF version.

Comments 3: (p<0.05), if attributable.

Response 3: Thank you very much for pointing out the problem. We apologize for not being able to express it clearly. We have made modifications in the corresponding positions and marked them in the PDF version.

Comments 4: The CFU (colony forming units) number. "After the two groups" - is not necessary.

Response 4: Thank you very much for pointing out the problem. We apologize for not being able to express it clearly. We have made modifications in the corresponding positions and marked them in the PDF version.

Comments 5: "were assessed"- is not necessary.

Response 5: Thank you very much for pointing out the problem. We have made changed the title into “The diversity and richness indices of bacterial and fungal communities in perennial poplar big trees and poplar seedlings in soil with continuous cropping.”

Comments 6: It is advisable to replace 0.51 with 1.96, changing numerator and denominator in a fraction, whien you calculate the ratio.

Response 6: Thank you very much for pointing out the problem. We apologize for not being able to express it clearly. We have made modifications in the corresponding positions and marked them in the PDF version.

Comments 7: Please specify here and throughout the text: whether the abundance of bacteria/fungi increased or decreased.

Response 7: Thank you very much for pointing out the problem. We’ve made changes in the corresponding positions.

Reviewer 2 Report

Comments and Suggestions for Authors

I  recommend accepting this manuscript for publication. The authors have made significant improvements that address all the reviewers' points and elevate the quality of the work. I  recommend accepting this manuscript for publication. 

Author Response

Thank you very much for taking the time to review this manuscript.

Reviewer 3 Report

Comments and Suggestions for Authors

The authors put a significant effort into modifying the manuscript.

I would suggest changing the title of Table 3. "The diversity and richness indices of bacterial and fungal communities were assessed
in the two soil treatments."  to "The diversity and richness indices of bacterial and fungal communities in perennial poplar big trees and poplar seedlings in soil with continuous cropping."
